# Prior Knowledge Integration via LLM Encoding and Pseudo Event Regulation for Video Moment Retrieval

## ABSTRACT

In this paper, we investigate the feasibility of leveraging large language models (LLMs) for integrating general knowledge and incorporating pseudo-events as priors for temporal content distribution in video moment retrieval (VMR) models. The motivation behind this study arises from the limitations of using LLMs as decoders for generating discrete textual descriptions, which hinders their direct application to continuous outputs like salience scores and inter-frame embeddings that capture inter-frame relations. To overcome these limitations, we propose utilizing LLM encoders instead of decoders. Through a feasibility study, we demonstrate that LLM encoders effectively refine inter-concept relations in multimodal embeddings, even without being trained on textual embeddings. We also show that the refinement capability of LLM encoders can be transferred to other embeddings, such as BLIP and T5, as long as these embeddings exhibit similar inter-concept similarity patterns to CLIP embeddings. We present a general framework for integrating LLM encoders into existing VMR architectures, specifically within the fusion module. The LLM encoder's ability to refine concept relation can help the model to achieve a balanced understanding of the foreground concepts (e.g., persons, faces) and background concepts (e.g., street, mountains) rather focusing only on the visually dominant foreground concepts. Additionally, we introduce the concept of pseudo-events, obtained through event detection techniques, to guide the prediction of moments within event boundaries instead of crossing them, which can effectively avoid the distractions from adjacent moments. The integration of semantic refinement using LLM encoders and pseudo-event regulation is designed as plug-in components that can be incorporated into existing VMR methods within the general framework. Through experimental validation, we demonstrate the effectiveness of our proposed methods by achieving state-of-the-art performance in VMR. The source code can be accessed at https://github.com/open_upon_acceptance.

## CCS CONCEPTS

• **Computing methodologies** → **Visual content-based indexing and retrieval**.

## KEYWORDS

Video Moment Retrieval, Highlight Detection, LLMs

## 1 INTRODUCTION

The rapid expansion of video content, driven by advancements in digital platforms and devices, has elevated video to become one of the most captivating and information-rich media formats today. However, this surge has also presented a significant challenge in efficiently navigating through vast amounts of video content to locate specific user-requested moments or highlights. In response to this challenge, research efforts have progressed from traditional approaches of moment retrieval (MR) and highlight detection (HD) to more advanced methods, including Moment-DETR [29], UMT [43], and UniVTG [36]. These cutting-edge techniques have pushed the boundaries in the field of video understanding, enabling more effective analysis and extraction of meaningful insights from videos.

An emerging trend observed in these advancements is the incorporation of prior knowledge into the learning process to enhance the semantic context and improve representation learning. This effort includes the expansion of datasets from a scale of hundreds [61] to tens of thousands [29], which improves the generality of resulting representations across broader domain. Another endeavor is the introduction of self-supervised pretriaining such as UniVTG [36] which has been conducted by integrating existing MR/HD datasets for in-domain knowledge incorporation and QVHignlights [29] which has been pretrained on Youtube subtitles for cross-domain knowledge embedding. While these have broken new ground for prior integration, the question of whether we can innovate beyond still remains open.

In a post-ChatGPT era, the inclination to utilize large language models (LLMs) arises naturally due to their success across various tasks. However, employing LLMs in MR/HD tasks proves to be challenging since our focus lies primarily on capturing fine-grained inter-frame salience, whereas LLMs excel in high-level semantic comprehension. In simpler terms, LLMs perform well in tasks such as captioning [56] or grounding [27], effectively describing the content of a video or image as a whole, but they lack the ability to compare the degree of semantic salience among individual frames. From a technical standpoint, most applications utilize LLMs as decoders, where visual representations are first converted into LLM-compatible tokens using models like Q-Former [32]. These transformed representations are then integrated into the context for generating outputs. Unfortunately, neither the Q-Former nor the LLM models have been trained with frame-level salience information during this process. Furthermore, the LLM decoders output textual tokens which are discrete and determinate and thus less compatible to the continuous and comparative salience scores. This also makes the customization of the decoders or integrating MLPs for fine-grained decision making less feasible.

In this paper, we tackle these limitations by utilizing LLM encoders instead of decoders. Through a feasibility study, we demonstrate that LLM encoders effectively refine inter-concept relations in multimodal embeddings, even when not trained on textual embeddings. We confirm that the refinement ability of LLM encoders can be transferred to other embeddings, such as BLIP [33] and T5 [57], as long as the inter-concept similarities exhibited by these embeddings demonstrate similar patterns to CLIP embeddings (which serve as the original input for most LLMs). Based on these findings, we propose a general framework for applying LLM encoders in VMR. We showcase that LLM encoders can be inserted into the fusion module of existing VMR architectures. By doing this, the LLM encoder's ability for inter-concept refinement can help the model

to have a comprehensive understanding of foreground concepts (e.g., *persons, faces*) and background concepts (e.g., *street, buildings*), to avoid the leaner to be misled by the visually dominant foreground concepts. This is especially important when those concepts are scattered over consecutive frames and the combination of their semantics cannot be identified without an inter-frame relation modeling. Furthermore, we introduce the use of pseudo-events, obtained through event detection techniques (e.g., [63]), as priors for the content distribution of videos. This approach guides the predicted moments to align with events rather than crossing unreasonable event boundaries, which can effectively avoid distractions from adjacent moments. The integrations of semantic refinement using LLM encoders and pseudo-event regulation are designed as plug-in components that can be incorporated into existing VMR methods constructed within the general framework. Through our experiments, we validate that these components can enhance the performance of five VMR frameworks: Moment-Detr [29], Uni-VTG [36], QD-DETR [50], CG-DETR [49], and EaTR [23].

## 2 RELATED WORK

### 2.1 Moment Retrieval and Highlight Detection

Moment Retrieval (MR) and Highlight Detection (HD) have become pivotal in navigating the growing expanse of video content. MR focuses on localizing video moments pertinent to textual descriptions, employing cross-modal interactions [45, 49, 50, 85, 87, 93] and temporal relation context [20, 89]. These tasks focus on localizing user-desired moments and scoring clip-wise correspondence to queries, respectively. Moment retrieval, aimed at retrieving user-specific video segments, has evolved significantly [10, 17, 20, 37, 65]. Traditional methods in this domain are categorized into proposal-based and proposal-free approaches. Proposal-Based Methods utilize predefined proposals like sliding windows [18, 20, 41, 85, 90]or temporal anchors [8, 39, 85, 86, 91], and in some cases, generate proposals [38, 62, 76, 79, 89]. The essence of these methods lies in matching these candidates with the text query. In contrast, proposal-free methods bypass the use of predefined candidates. Instead, they focus on encoding multimodal knowledge and directly predict temporal spans using regression heads, making the process more streamlined and potentially more accurate.

Highlight detection, on the other hand, concentrates on scoring the importance of each video clip, whether based solely on visual [3, 60, 77, 80] or combined visual-audio inputs [3, 19, 21, 77]. The methods here vary in their approach to label granularity, with supervised [19, 68, 80], weakly supervised [5, 53, 77][6, 49, 68], and unsupervised methods [3, 25, 47, 60] all contributing to the field. The introduction of the QVHighlights dataset [29] marked a significant shift in this domain, prompting a combined consideration of these problems. Emerging approaches post-QVHighlights adopt either DETR or regression-based frameworks. For instance, UMT [43] explores additional audio modalities, while QD-DETR [50] and CG-DETR [49]innovate upon the DETR architecture. Other studies [36, 82] underscore the importance of pretraining.

### 2.2 Vision-Language Models

Inspired by the success of ChatGPT, significant efforts have been dedicated to developing visual-language models (VLMs) based on LLMs [11, 12, 51, 52, 70–72]. One commonly employed approach involves encoding images into tokens that are compatible with LLM inputs. The pretrained LLMs then serve as decoders for generating textual descriptions. A well-known framework is BLIP-2 [32], which employs QFormer as the encoder and LLMs like FlanT5 [13] as the decoder. Several successful examples have adopted this framework. For instance, MiniGPT4 [94] replaces the LLM with a more advanced model called LLaMA [72]. InstructBLIP [14] fine-tunes the model using high-quality instructions, while LLaV [40] explores CLIP's open-set visual encoder connected to Vicuna's linguistic decoder and performs end-to-end fine-tuning on the generated instructional vision-language data. Similar frameworks have been extended to video understanding, resulting in models like VideoChat [34], Video-ChatGPT [46], and LLaMA-VID [35].

Other variants that may not strictly stick to the framework include GILL [26], Emu [69], mPLUG [31], CogVLM [75] and MiniGPT-5 [92] for visual question answering, and Vision-LLM [74], Kosmos-2 [55], Qwen-VL [4], MiniGPT-v2 [9], GPT4-Vison [1], and GeminiProVision [70] that support multimodal inputs and outputs. However, the role of LLMs in these frameworks is still a decoder for discrete outputs (mainly textual descriptions/answers), which makes them limited to continuous outputs such as the salience scores or inter-frame correlations. In this paper, we explore the feasibility of employing LLMs as encoders for semantic relation refinement, based on which the outputs are still continuous embeddings and thus open the opportunity for fine-grained information processing or decoding. This particular direction remains under-explored, with only one existing work [54] in the literature that has attempted a similar approach for image understanding tasks. This paper distinguishes from [54] in two aspects: we conduct a feasibility study to investigate the rationale behind this approach, and we focus on the integration of prior knowledge in the LLM.

## 3 LEVERAGING LLM ENCODERS FOR INTER-CONCEPT RELATION REFINEMENT

Before introducing our VMR framework, let us examine the viability of utilizing LLM encoders to refine inter-concept relations. LLM encoders serve as natural inter-concept relation refiners since they operate as Transformers, taking concepts (represented as token embeddings) as input and generating refined embeddings as output. This becomes particularly evident when contrastive losses are employed, as they encourage similar concepts to move closer in the embedding space while pushing dissimilar ones further apart. However, this approach appears to be effective only for textual embeddings and is tightly coupled with the specific embeddings used by the LLM (e.g., CLIP [56], BLIP [33], T5 [57]). In this feasibility study, our aim is to investigate whether this approach can be applied to multimodal embeddings and delve into the underlying reasoning. Additionally, we will explore the possibility of reducing computational costs by utilizing only a subset of internal layers from an LLM encoder instead of loading the entire model.

### 3.1 LLM Encoders are Relation Refiners

We conducted an experiment to validate the hypothesis that an LLM Encoder can be utilized for relation refinement. The experiment commenced by requiring GPT-4 to recommend 1,000 triplets,

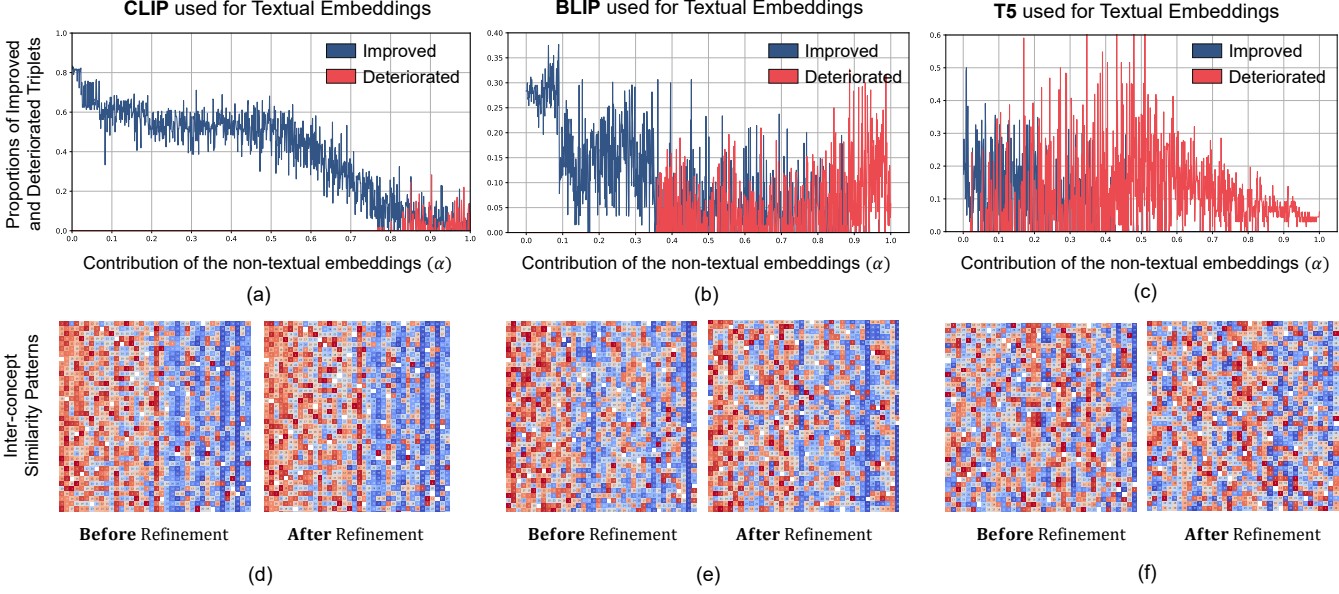

**Figure 1: The proportions of improved and deteriorated triplets after the refinement (a–c), and the inter-concept similarity matrices of the concept embeddings before and after the refinement (d–f): in (a) and (d), CLIP is used as the textual embeddings, while BLIP is used for (b) and (e), and T5 is used for (c) and (f).**

wherein each triplet (e.g., *sock, shoe, galaxy*) comprises two "paired" concepts (e.g., *sock* and *shoe*) that possess a strong semantic relation, while the third concept (e.g., *galaxy*) is significantly less related to the other two. We label a triplet as "unreasonable" if the similarity between the paired concepts is lower than that of any unpaired concepts; otherwise, it is deemed "reasonable." We input CLIP embeddings of these concepts into the LLaMA-2 [72] encoder and obtain refined embeddings. To assess the extent of refinement in the embeddings, we measure the percentage of unreasonable triplets in the CLIP space that become reasonable in the refined space. The result shows that 83.33% (300/360) of the initially unreasonable triplets have been refined to a reasonable state, while none of the reasonable triplets were transformed into unreasonable ones.

## 3.2 LLM Encoders for Multimodal Embeddings

While the LLM encoder's capability to refine semantic relations is expected, our primary focus lies in investigating whether this ability can be transferred to multimodal embeddings. In most applications, multimodal embeddings are generated by combining non-textual embeddings with textual ones using methods such as weighted summation or more advanced cross-attention mechanisms. These fusion approaches aim to align the non-textual embeddings with the textual ones, forming the foundation for stacking and fine-tuning task-specific decision-making layers. In our subsequent experiment, we simulate the fusion and alignment process by fusing the textual embeddings with randomly generated embeddings.

**Dose the LLM encoders work for fused embeddings?** Let $\mathbf{c}$ represent a textual concept embedding obtained through CLIP, and let $\mathbf{c}'$ denote a concept-dependent randomly generated vector used to simulate a non-textual embedding. We combine these two

embeddings using a weighted summation, as

$$\mathbf{c}^* = (1 - \alpha)\mathbf{c} + \alpha\mathbf{c}'. \tag{1}$$

It should be noted that the simulated embedding vector $\mathbf{c}'$ is specific to each concept. This implies that each concept is associated with its own fixed vector, and the vectors corresponding to different concepts are distinct from one another. Due to the random nature of the vectors $\mathbf{c}'$, the expected similarities between such non-textual embeddings are zero. This allows $\mathbf{c}'$ to simulate real-world scenarios where non-textual embeddings are extracted using specific encoders. In these scenarios, the embeddings are aligned to concepts, but the inter-concept similarities are not necessarily regulated in a reasonable manner. Consequently, the parameter $\alpha$ in Eq. (1) controls the extent to which the textual embeddings are influenced by the randomness and misalignment introduced by the non-textual embeddings.

The results in Fig. 1(a) indicate that when the textual embeddings are predominant (i.e., $\alpha \leq 0.5$), the LLaMA encoder effectively refines the majority of unreasonable pairs without causing any deterioration. Fortunately, this aligns perfectly with the requirements of many multimodal applications, where, during the representation learning phase, the goal is to align the non-textual modalities with the textual one.

**What will happen when non-textual embeddings are distorted?** To expand the experiment, we introduce a distortion factor by randomly setting elements of a non-textual embedding vector $\mathbf{c}'$ to zero with a probability of $p$, which results in a distorted vector $\mathbf{c}'_p$. This simulation reflects real-world multimodal scenarios where the alignment of non-textual embeddings to concepts is not consistently reliable. We can substitute the $\mathbf{c}'$ with $\mathbf{c}'_p$ in Eq. (1) and repeat the experiment. The results shown in Fig. 2 demonstrate

that the distortion of non-textual embeddings would not degrade the refinement when the textual embeddings are dominant.

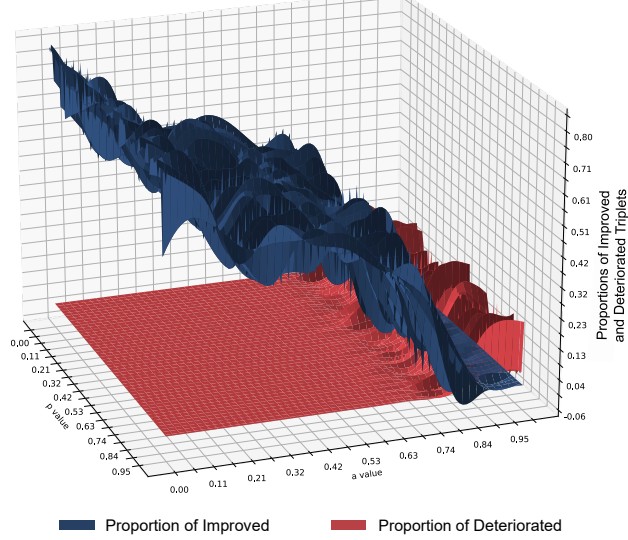

**Figure 2: Proportions of Improved and Deteriorated Triplets over the contribution of non-textual embeddings controlled by the $\alpha$ and the degree of alignment between the textual and non-textual embeddings controlled by the distortion probability $p$.**

Through the conducted experiments, we have confirmed the effectiveness of LLM encoders as reliable relation refiners. This can be attributed to the fact that LLM encoders are based on the Transformer architecture. Transforms heavily rely on self-attention mechanisms which are driven by the inter-concept (token) similarities. In other words, as long as the input concepts maintain inter-concept similarities similar to those of the CLIP embeddings, the LLM encoder can refine the concept relations in a manner comparable to using CLIP embeddings as input. In the experiment above, when there are two CLIP concept embeddings $\mathbf{c}_1$ and $\mathbf{c}_2$, the similarity of their fused embeddings is written

$$sim(\mathbf{c}_1^*, \mathbf{c}_1^*) = \mathbf{c}_1^*(\mathbf{c}_2^*)^\top$$
$$= ((1-\alpha)\mathbf{c}_1 + \alpha\mathbf{c}_1')((1-\alpha)\mathbf{c}_2 + \alpha\mathbf{c}_2')^\top \quad (2)$$
$$= (1-\alpha)^2\mathbf{c}_1\mathbf{c}_2^\top + (1-\alpha)\alpha(\mathbf{c}_1\mathbf{c}_2'^\top + \mathbf{c}_1'\mathbf{c}_2^\top) + \alpha\mathbf{c}_1'\mathbf{c}_2'^\top.$$

Obviously, the expectations of concept similarities before and after fusion have a relation of $\mathbb{E}[\mathbf{c}_1^*(\mathbf{c}_2^*)^\top] \approx (1-\alpha)^2\mathbb{E}[\mathbf{c}_1(\mathbf{c}_2)^\top]$. This is because the expectations of the last two terms in Eq. (2) are zero due to the randomness of the simulated non-textual embeddings. This indicates that the fused concept embedding still maintains similar (even when scaled) inter-concept similarities compared to the original embeddings, which explains why the relations of the fused concepts can also be refined using the LLM encoder.

To validate the discovery, we propose replacing the CLIP embeddings with its variant, BLIP, and introducing a more distant textual embedding model, T5. The results depicted in Fig. 1(b) and Fig. 1(c) indicate improvements when either BLIP or T5 embeddings are

utilized. It is not surprising that by using BLIP embeddings which are more similar to CLIP embeddings, the results demonstrate a better balance between the improved and deteriorated, and the pattern closely resembles that of the CLIP. However, T5 still works because it is also a well-trained model in the text domain, in which we expect a significant overlap in inter-concept similarities with CLIP. In Fig. 1(d–f), we visualize the inter-concept similarities of these three types of embeddings before and after the refinement. It is evident that the patterns exhibit similarity, and the refinement process further enhances this similarity.

### 3.3 Using A Subset of Layers as the Encoder

Our last experiment for the feasibility is to use a subset of LLaMA encoder as a relation refiner. The motivation is that the layers of a Transformer process data in a similar manner, suggesting that a subset of layers may possess a similar ability to refine relations. If this hypothesis holds true, it would enable a significant reduction in computational requirements, thereby enhancing the feasibility of applying this approach to a wider range of applications. The results in Fig. 3 validate the hypothesis by demonstrating that a subset of the LLaMA encoder can also effectively refine relations.

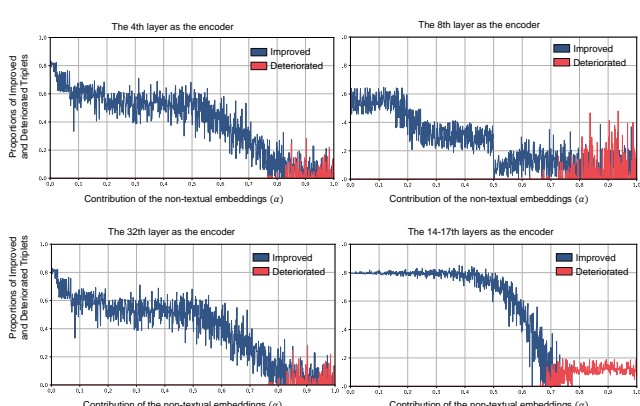

**Figure 3: The impact of utilizing specific layers from the LLM encoder for relation refinement. The performance of individual layers ($4^{th}$, $8^{th}$, and $32^{nd}$) as well as combined layers ($14^{th}$ to $17^{th}$) have been studied.**

## 4 METHOD

### 4.1 A General Framework for VMR

Before introducing the proposed method, we summarize existing VMR approaches into a general framework to ease the description (see Fig. 4). The VMR task takes a video $V$ and a textual query $Q$ as input and predicts a set of candidate video segments $\{\mathbf{m}_k\}$ that are relevant to the query $Q$. This process is accomplished by an inference process of a neural network $f_\theta$ with parameters $\theta$ as

$$\{\mathbf{m}_k\} = f_\theta(Q, V). \quad (3)$$

The process begins by encoding the query and video into embeddings $\{\mathbf{q}_i\}$ and $\{\mathbf{v}_i\}$, respectively, as

$$\{\mathbf{q}_i\} = CLIP_t(Q), \ \{\mathbf{v}_j\} = CLIP_v(V) \oplus SlowFast(V), \quad (4)$$

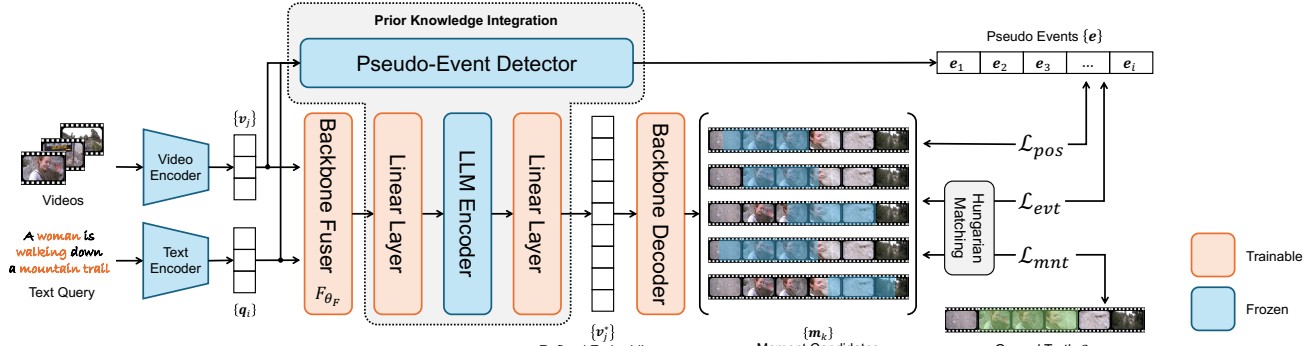

**Figure 4: The proposed general framework for VMR with the proposed prior knowledge integration components.**

where the $CLIP_t$ and $CLIP_v$ are the textual and visual encoders of CLIP [56], respectively, and the SlowFast [16] is a commonly used video encoder. The operator $\oplus$ denotes concatenation. The embeddings are then fused by a network $F$ with parameters $\theta_F$ as

$$\{\mathbf{v}_j^*\} = F_{\theta_F}(\{\mathbf{q}_i\}, \{\mathbf{v}_j\}). \qquad (5)$$

A decoder $D$ with parameters $\theta_D$ then predicts the moments as

$$\{\mathbf{m}_k\} = D_{\theta_D}(\{\mathbf{v}_j^*\}). \qquad (6)$$

A loss function is defined with the ground truth $G$ as

$$\mathcal{L}_{mnt} = Dist(\{\mathbf{m}_k\}, G), \qquad (7)$$

where $Dist$ is a distance metric and often implemented using Hungarian Match [6]. The learning goal is to find an optimal set of parameters $\theta = \{\theta_F, \theta_D\}$.

A large portion of VMR methods can be structured within this general framework, with their unique characteristics stemming from proposed variations in the fusion network $Fu$ or adjustments to the loss function $\mathcal{L}$. In our paper, we introduce two plug-in elements to this framework, which implement prior integration of semantic refinement and pseudo-event regulation, respectively. These plug-in elements are designed to be compatible with a range of existing methods, emphasizing their versatility and adaptability.

## 4.2 Integration of Semantic Refinement

To incorporate semantic refinement within the general framework, we can insert an LLM encoder between the fusion and decoding processes. The feasibility is that the fusion process exhibits similarities to the process described in Eq. (1), where video embeddings are merged with textual query embeddings. Furthermore, CLIP embeddings play a significant role in both the encoding and fusion processes, making them suitable as inputs of the LLM encoder, as observed in our feasibility study. Considering the rationale behind it, the utilization of the LLM encoder for inter-concept refinement can enhance the model's overall comprehension of both foreground concepts (e.g., *persons, faces*) and background concepts. This is crucial in preventing the model from being misled by visually dominant foreground concepts. This becomes particularly significant when these concepts are dispersed across consecutive frames, and their combined semantics cannot be identified without proper modeling

of inter-frame relations. However, to adapt to the input dimension of the LLM encoder, we need to include a linear layer before and after the LLM encoder. The integration is then written by replacing Eq. (5) as

$$\{\mathbf{v}_j^*\} = L2_{\theta_{L2}}\left[\mathbf{LLM}\left(L1_{\theta_{L1}}\left[F_{\theta_F}(\{\mathbf{q}_i\}, \{\mathbf{v}_j\})\right]\right)\right], \qquad (8)$$

where the $L1$ and $L2$ are the two linear adapter layers with parameters $\theta_{L1}$ and $\theta_{L2}$, respectively. In order to improve computational efficiency, we can select a subset of layers from the LLM to serve as the refiner.

## 4.3 Integration of Pseudo-Event Regulation

The motivation for this regulation arises from the recognition that valid moments should remain within the boundaries of events instead of crossing them. By guiding the predicted moments to align with the content distribution indicated by the event boundaries, we can effectively eliminate distractions from adjacent irrelevant moments. To this end, we utilize event detectors such as the recursive TSM parsing mechanism in UBoCo [24] to generate pseudo events for a given video. These pseudo events serve as a prior for the distribution of event boundaries. This prior can be used to guide the predicted moments into positions between event boundaries by penalizing those that extend beyond the boundaries. Let us denote the position and with of a predicted moment $\mathbf{m}_k$ as a vector $\mathbf{p}_k = [pos_k, wid_k]$ and the detected pseudo events as a set $\mathbf{e} = [pos_e, wid_e]$. This can be implemented by introducing a pseudo-event regulated loss as

$$\mathcal{L}_{evt} = \sum_{\mathbf{e}\in\{\mathbf{e}\}}\left(\lambda_{L1}\|\mathbf{e} - \mathbf{m}_k\|_1 + \lambda_{IoU}\mathcal{L}_{IoU}(\mathbf{e}, \mathbf{m}_k)\right), \qquad (9)$$

where $\mathcal{L}_{IoU}$ is the widely adopted Generalized Intersection Over Union (GIoU) loss [59], $\lambda_{L1}$ and $\lambda_{IoU}$ are the weights for the L1 norm and GIoU loss, respectively.

The aforementioned regulation is designed to occur after the prediction process, as a post-validation step. We have also devised a pre-reinforcement-based regulation technique to further enhance the alignment between predicted moments and events. The concept behind this technique involves adjusting the position embeddings of

frames by reinforcing the notion that frames within the same event should possess similar position embeddings. By doing so, the prediction process is encouraged to select frames from the same event for moment composition, rather than across different events. We set the target position embeddings as those of the fused embeddings after $F_{\theta_F}(\cdot)$. For an event $\mathbf{e}$, we denote the position embeddings of its member frames as a matrix $\mathbf{P}_e$, wherein the position embedding of the center frame is $\mathbf{p}_e$. The position regulation is then written as

$$\mathcal{L}_{pos} = \sum_{\mathbf{e} \in \{\mathbf{e}\}} \exp\left[\frac{1}{N}\mathbf{1}^{\top}\mathrm{abs}\left(\mathbf{P}_e - \mathbf{p}_e \cdot \mathbf{1}^{\top}\right)\mathbf{1}\right], \quad (10)$$

where $\mathbf{1}$ is a vector of ones. The $\mathcal{L}_{pos}$ encourages the position embeddings of the member frames to that of the center embedding within an event.

By incorporating both semantic refinement and pseudo-event regulation, the loss function is expanded as

$$\mathcal{L} = \mathcal{L}_{mnt} + \lambda_e \mathcal{L}_{evt} + \lambda_p \mathcal{L}_{pos}, \quad (11)$$

where $\lambda_e$ and $\lambda_p$ are hydrometers to balance the learning. The extended set of parameters to be tuned is denoted as $\theta = \{\theta_F, \theta_D, \theta_{L1}, \theta_{L2}\}$.

## 5 EXPERIMENTS

### 5.1 Dataset and Evaluation Metrics

To evaluate the performance of our proposed method, we conduct extensive experiments across multiple tasks, organized into three panels: 1) *Joint moment retrieval and highlight detection* on QVHighlights [29], which includes more than 10,000 videos with high-quality text queries. For moment retrieval, we report Recall@1 with IoU thresholds of 0.5 and 0.7, mean Advance Precision (mAP) with IoU thresholds of 0.5 and 0.75, and mean mAP across a range of IoU thresholds [0.5:0.05:0.95]. For highlight detection, we report mAP and HIT@1, which consider a segment as truly "positive" when its predicted saliency score is rated as "very good". 2) *Individual moment retrieval* on Charades-STA [17] and TACoS [58]. The metrics involve Recall@1 and Recall@5 with IoU thresholds of 0.5 and 0.7. 3) *Highlight detection* on TVSum [67] and Youtube-HL [68], where Top-5 mAP and mAP are employed as performance measures, respectively. All metrics are aligned with previous studies [29, 43, 50].

### 5.2 Implementation Details

We utilize the SlowFast [16] and CLIP [56] models to extract video features on QVHighlights dataset every 2 seconds. We adopt Slowfast+Clip and VGG [64] on Charades-STA and TACoS datasets, where video features are extracted every 1 second and 2 seconds, respectively. For YouTube-HL and TVSum datasets, we extract clip-level features by a pre-trained I3D [7]. Following previous methods [43], each feature vector captures 32 consecutive frames and is considered as a clip when the overlap exceeds 50% with each other.

We utilize the 14th to 17th encoding layers of LLaMA (7B) across all configurations. The feature fusion encoder $F$ and decoder $D$ are implemented using 6-layer Transformer. The loss balancing hydrometers $\lambda_e$ and $\lambda_p$ are set to 0.1 and 0.001, respectively. We use AdamW [44] with a weight decay of 1e-5 and apply an additional pre-discard rate of 0.5 to the visual inputs. We train our model across five datasets with specific settings for each: 1) QVHighlights:

**Table 1: Results of joint moment retrieval and highlight detection (HD) on QVHighlights test split [28].**

| Method | Moment Retrieval | | | | | HD | |
| | R1 | | mAP | | | ≥ Very Good | |
| | @ 0.5 | @ 0.7 | @ 0.5 | @ 0.75 | Avg. | mAP | HIT@1 |
|---|---|---|---|---|---|---|---|
| BeautyThumb [66] | - | - | - | - | - | 14.36 | 20.88 |
| DVSE [42] | - | - | - | - | - | 18.75 | 21.79 |
| MCN [20] | 11.41 | 2.72 | 24.94 | 8.22 | 10.67 | - | - |
| CAL [15] | 25.49 | 11.54 | 23.40 | 7.65 | 9.89 | - | - |
| CLIP [56] | 16.88 | 5.19 | 18.11 | 7.0 | 7.67 | 31.30 | 61.04 |
| XML [30] | 41.83 | 30.35 | 44.63 | 31.73 | 32.14 | 34.49 | 55.25 |
| XML+ [30] | 46.69 | 33.46 | 47.89 | 34.67 | 34.90 | 35.38 | 55.06 |
| Moment-DETR [29] | 52.89 | 33.02 | 54.82 | 29.40 | 30.73 | 35.69 | 55.60 |
| UMT [43] | 56.23 | 41.18 | 53.83 | 37.01 | 36.12 | 38.18 | 59.99 |
| UniVTG [36] | 58.86 | 40.86 | 57.60 | 35.59 | 35.47 | 38.20 | 60.96 |
| QD-DETR [49] | 62.40 | 44.98 | 62.52 | 39.88 | 39.86 | 38.94 | 62.40 |
| CG-DETR [50] | 65.43 | 48.38 | 64.51 | 42.77 | 42.86 | 40.33 | **66.21** |
| Ours | **66.73** | **49.94** | **65.76** | **43.91** | **44.05** | **40.33** | 65.69 |

Learning rate of 1e-4, batch size of 32, for 200 epochs. 2) Charades-STA: Learning rate of 2e-4, batch size of 32, for 200 epochs. 3) TACoS: Learning rate of 2e-4, batch size of 16, for 200 epochs. 4) YouTube-HL: Learning rate of 1e-4, batch size of 4, for 2,000 epochs. 5) TVSum: Learning rate of 1e-4, batch size of 1, for 2,000 epochs.

### 5.3 Comparison with State-of-the-arts

*Joint Moment Retrieval and Highlight Detection.* Our evaluation is detailed in Tab. 1, specifically focusing on the QVHighlights test split. It is obvious that the proposed approach reaches the highest scores across almost all evaluation metrics. Specifically, we achieve an average mAP of 44.05 on moment retrieval and a HIT@1 of 65.69 on highlight detection, significantly surpassing traditional methods. When compared to contemporary models such as CG-DETR, our method still exhibits clear advantages, achieving superior mAP performance in terms of 2.78% on moment retrieval. This demonstrates the effectiveness of our approach in handling complex video analysis tasks.

*Moment Retrieval.* As shown in Tab. 2, we report the results of moment retrieval on two additional benchmarks, TACoS and Charades-STA. Notably, our approach outperforms all SOTA methods even without pre-training. Specifically, we exceed other methods by margins of 0.19% to 34.28% on TACoS and 0.6% to 21.82% on Charades-STA in terms of mIoU. The consistent outperformance across various benchmarks and comparison with a wide range of SOTA methods suggest the effectiveness of incorporating prior knowledge from large language models.

*Highlight Detection.* We conduct more experiments on highlight detection. Tab. 3 displays the Top-5 mAP on the TV-Sum dataset, while Tab. 4 details the mAP performance on YouTube-HL. From an overall perspective, our method demonstrates significant improvements, achieving an mAP of 88.1% on TV-Sum and 75.3% on YouTube-HL.

**Table 2: Moment retrieval results tested on TACoS and Charades-STA datasets. Video features are extracted using Slowfast and CLIP.**

| Method | TACoS | | | Charades-STA | | |
|--------|-------|-------|------|--------------|-------|------|
| | R@0.3 | R@0.7 | mIoU | R@0.3 | R@0.7 | mIoU |
| 2D-TAN | 40.01 | 12.92 | 27.22 | 58.76 | 27.50 | 41.25 |
| VSLNet | 35.54 | 13.15 | 24.99 | 60.30 | 24.14 | 41.58 |
| Moment-DETR | 37.97 | 11.97 | 25.49 | 65.83 | 30.59 | 45.54 |
| QD-DETR | - | - | - | - | 32.55 | - |
| LLaViLo | - | - | - | - | 33.43 | - |
| UniVTG | 51.44 | 17.35 | 33.60 | 70.81 | 35.65 | 50.10 |
| CG-DETR | 52.23 | 22.23 | 36.48 | 70.43 | 36.34 | 50.13 |
| Ours | **52.73** | **22.78** | **36.55** | **70.91** | 36.49 | **50.25** |

**Table 3: Highlight detection results on TV-Sum, † denotes the methods that utilize the audio modality.**

| Method | VT | VU | GA | MS | PK | PR | FM | BK | BT | DS | Avg. |
|--------|-----|-----|-----|-----|-----|-----|-----|-----|-----|-----|------|
| sLSTM [88] | 41.1 | 46.2 | 46.3 | 47.7 | 44.8 | 46.1 | 45.2 | 40.6 | 47.1 | 45.5 | 45.1 |
| SG [48] | 42.3 | 47.2 | 47.5 | 48.9 | 45.6 | 47.3 | 46.4 | 41.7 | 48.3 | 46.6 | 46.2 |
| LIM-S [78] | 55.9 | 42.9 | 61.2 | 54.0 | 60.3 | 47.5 | 43.2 | 66.3 | 69.1 | 62.6 | 56.3 |
| Trailer [73] | 61.3 | 54.6 | 65.7 | 60.8 | 59.1 | 70.1 | 58.2 | 64.7 | 65.6 | 68.1 | 62.8 |
| SL-Module [81] | 86.5 | 68.7 | 74.9 | 86.2 | 79.0 | 63.2 | 58.9 | 72.6 | 78.9 | 64.0 | 73.3 |
| QD-DETR [50] | 88.2 | 87.4 | 85.6 | 85.0 | 85.8 | 86.9 | 76.4 | 91.3 | 89.2 | 73.7 | 85.0 |
| UniVTG [36] | 83.9 | 85.1 | 89.0 | 80.1 | 84.6 | 81.4 | 70.9 | 91.7 | 73.5 | 69.3 | 81.0 |
| MINI-Net† [22] | 80.6 | 68.3 | 78.2 | 81.8 | 78.1 | 65.8 | 57.8 | 75.0 | 80.2 | 65.5 | 73.2 |
| TCG† [84] | 85.0 | 71.4 | 81.9 | 78.6 | 80.2 | 75.5 | 71.6 | 77.3 | 78.6 | 68.1 | 76.8 |
| Joint-VA† [2] | 83.7 | 57.3 | 78.5 | 86.1 | 80.1 | 69.2 | 70.0 | 73.0 | **97.4** | 67.5 | 76.3 |
| UMT† [43] | 87.5 | 81.5 | 88.2 | 78.8 | 81.4 | 87.0 | 76.0 | 86.9 | 84.4 | **79.6** | 83.1 |
| Ours | **90.8** | **91.9** | **94.2** | **88.7** | **85.8** | **90.4** | **78.6** | **93.4** | 88.3 | 78.7 | **88.1** |

**Table 4: Performance of mAP for highlight detection on YouTube-HL. † denotes using audio modality.**

| Method | Dog | Gym. | Par. | Ska. | Ski. | Sur. | Avg. |
|--------|------|------|------|------|------|------|------|
| RRAE [83] | 49.0 | 35.0 | 50.0 | 25.0 | 22.0 | 49.0 | 38.3 |
| GIFs [19] | 30.8 | 33.5 | 54.0 | 55.4 | 32.8 | 54.1 | 46.4 |
| LSVM [68] | 60.0 | 41.0 | 61.0 | 62.0 | 36.0 | 61.0 | 53.6 |
| LIM-S [78] | 57.9 | 41.7 | 67.0 | 57.8 | 48.6 | 65.1 | 56.4 |
| SL-Module [81] | 70.8 | 53.2 | 77.2 | 72.5 | 66.1 | 76.2 | 69.3 |
| QD-DETR [50] | 72.2 | **77.4** | 71.0 | 72.7 | 72.8 | 80.6 | 74.4 |
| UniVTG [36] | 71.8 | 76.5 | 73.9 | 73.3 | 73.2 | 82.2 | 75.2 |
| MINI-Net† [22] | 58.2 | 61.7 | 70.2 | 72.2 | 58.7 | 65.1 | 64.4 |
| TCG † [84] | 55.4 | 62.7 | 70.9 | 69.1 | 60.1 | 59.8 | 63.0 |
| Joint-VA† [2] | 64.5 | 71.9 | 80.8 | 62.0 | 73.2 | 78.3 | 71.8 |
| UMT † [43] | 65.9 | 75.2 | **81.6** | 71.8 | 72.3 | **82.7** | 74.9 |
| Ours | **73.6** | 74.2 | 72.5 | **75.3** | **73.4** | 82.5 | **75.3** |

## 5.4 Ablation study

Due to the space limitation, we limit the ablation study on QVHigh-lights validation split. QD-DETR [50] is temporally used as the baseline, and later we will conduct a compatibility study to show that our proposed plug-in components can be used by other 5 different frameworks to improve their performances. The results are shown in Tab. 5, where a consistent performance gain is observed

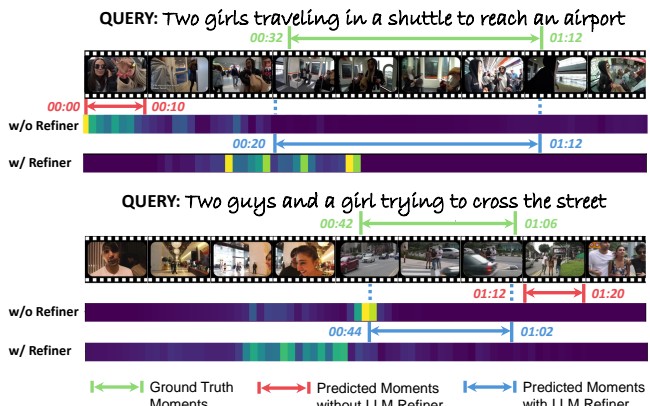

**Figure 5: Illustration of the effectiveness of using the LLM as a relation refiner. The predictions with the LLM encoder are better aligned with the ground truth. The model without the refiner focuses more on the visually dominate concepts (e.g., girls, guys), while with the refiner, contextual concepts (e.g., traveling, crossing-street) can be further incorporated.**

**Table 5: Ablation study on QVHighlights validation split.**

| | Models | R1@0.5 | R1@0.7 | mAP |
|---|--------|--------|--------|-----|
| 1 | Baseline | 63.87 | 48.71 | 41.46 |
| 2 | Baseline + LLM | $66.71_{2.8\uparrow}$ | $49.42_{0.7\uparrow}$ | $45.69_{4.2\uparrow}$ |
| 3 | Baseline + $\mathcal{L}_{\text{evt}}$ | $65.16_{1.3\uparrow}$ | $50.26_{1.6\uparrow}$ | $45.47_{4.0\uparrow}$ |
| 4 | Baseline + $\mathcal{L}_{\text{pos}}$ | $64.58_{0.7\uparrow}$ | $49.48_{0.8\uparrow}$ | $43.88_{2.4\uparrow}$ |
| 5 | Baseline + LLM + $\mathcal{L}_{\text{evt}}$ | $66.00_{2.1\uparrow}$ | $51.55_{2.8\uparrow}$ | $45.78_{4.3\uparrow}$ |
| 6 | Baseline + LLM + $\mathcal{L}_{\text{evt}}$ + $\mathcal{L}_{\text{pos}}$ | $66.58_{2.7\uparrow}$ | $51.10_{2.4\uparrow}$ | $46.24_{4.8\uparrow}$ |

when each component is combined into the framework, resulting in an improvement ranging from 0.7% to 4.8% in mAP.

***Effectiveness of using LLM encoders as relation refiners:*** It is evident in Tab. 5 that LLM can bring further improvement gain due to its ability to refine the inter-frame relation. To grasp more insights, the predicted moments with and without the LLM refiner are shown in Fig. 5 on two exemplar video segments. In the example of the "*two girls traveling in a shuttle*", the model without the LLM refiner focuses more on a moment that the *two girls* are visually dominant in the video, while the model with the LLM refiner focuses on a moment that the concepts of *girls, traveling* and, *shuttle-to-airport* all appeared with comparably less but equivalent salience in the video. This is an indication that the LLM refiner can help model the inter-concept relation in a better way. Similarly, in the second example, the model without the refiner focuses more on the concepts of *two-guy* and *girl*, and with the LLM refiner, the presence of the action of *cross-street* has been further considered.

***Effectiveness of the pseudo-event regulation:*** In Fig. 6, we give two examples to illustrate the effectiveness. It is evident that the predicted moments align with the pseudo-event boundaries in a better way, eliminating the distractions from adjacent moments.

***Effectiveness of position embedding regulation:*** After implementing pseudo-events that identify event boundaries, we can use these boundaries to refine the distribution of position embeddings.

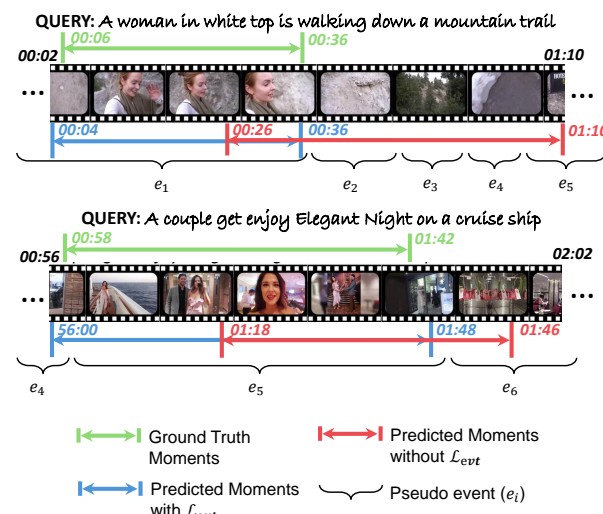

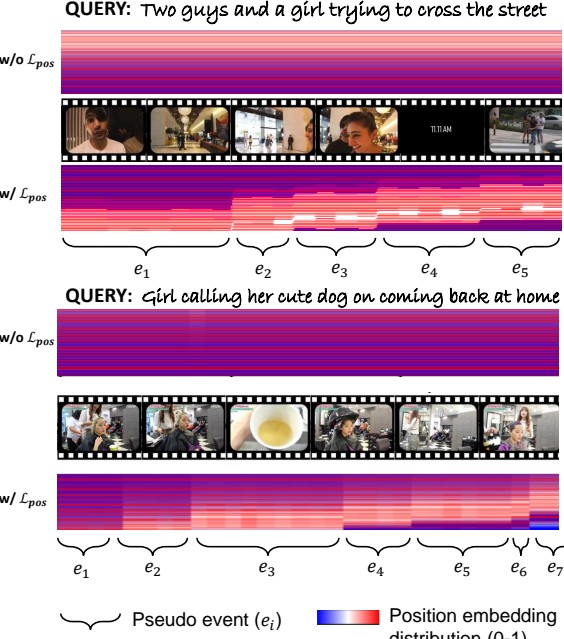

**Figure 6: Illustration of the effectiveness of the event regulation. The regulation guides predicted moments to reside inside the event boundaries rather than crossing them, which eliminates the distractions from adjacent moments.**

This adjustment aims for the position embeddings to align more closely with the event distribution in the video, as illustrated in Fig. 7. The impact of adopting $\mathcal{L}_{pos}$ is distinctly visible: prior to its use, the distribution of position embeddings is notably dispersed, while with the regulated loss $\mathcal{L}_{pos}$, the position embeddings show better alignment to the event boundaries.

***Study of the compatibility to different frameworks:*** By inserting the proposed integration components into various VMR frameworks, in Tab. 6, our method demonstrates high adaptability and compatibility with existing frameworks, indicated by a consistent performance gain over the original settings. The improvements observed are not merely instances of isolated success but reflect consistent effectiveness for the proposed integration components.

***Qualitative Results:*** Due the space limitation, we can study only a few examples in this section. We have included more illustrations and qualitative analysis in the Appendix.

## 6 CONCLUSION

In conclusion, our work presents an approach for enhancing video moment retrieval (VMR) by integrating large language model (LLM) encoders and pseudo-event regulation. The LLM encoders contribute to refining multimodal embeddings and their inter-concept relations, successfully applied to various embeddings such as CLIP and BLIP. We also use pseudo-events as temporal content distribution priors that aid in aligning moment predictions with actual event boundaries, addressing a previously underexplored aspect of VMR. The proposed methods serve as plug-in components, compatible with existing VMR frameworks, and have been empirically validated to achieve state-of-the-art performance. This study not only addresses the challenges posed by the vast and growing video content but also opens new avenues for the application of LLMs in fine-grained video analysis tasks.

**Figure 7: Illustration of the effectiveness of the position embedding regulation. The resulting embeddings align with the event distribution in a better way.**

**Table 6: Performance comparison of our integrated method with existing VMR frameworks (base models). Please note that the reported results of the base models might slightly differ from those in the original papers due to variations in the environment configuration. We have utilized the released source code, but the exact replication of the original environment is not feasible.**

| Method | Metric | Original | Integrated | Gain (%) |
|---|---|---|---|---|
| Moment-DETR [29] | MR-full-R1@0.5 | 53.90 | 59.23 | +9.89 |
| | MR-full-R1@0.7 | 34.80 | 38.52 | +10.69 |
| | MR-full-mAP | 32.20 | 34.36 | +6.71 |
| QD-DETR [50] | MR-full-R1@0.5 | 62.70 | 62.90 | +0.32 |
| | MR-full-R1@0.7 | 46.70 | 46.45 | -0.54 |
| | MR-full-mAP | 41.20 | 41.34 | +0.34 |
| EaTR [23] | MR-full-R1@0.5 | 57.42 | 63.03 | +9.77 |
| | MR-full-R1@0.7 | 42.58 | 46.06 | +8.17 |
| | MR-full-mAP | 38.98 | 41.05 | +5.31 |
| UniVTG [36] | MR-full-R1@0.5 | 59.74 | 59.70 | -0.07 |
| | MR-full-R1@0.7 | 35.59 | 38.84 | +9.13 |
| | MR-full-mAP | 36.13 | 36.36 | +0.64 |
| CG-DETR [49] | MR-full-R1@0.5 | 67.40 | 70.26 | +4.24 |
| | MR-full-R1@0.7 | 52.10 | 53.61 | +2.90 |
| | MR-full-mAP | 44.90 | 45.78 | +1.96 |

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
