# OpenReview forum: "Prior Knowledge Integration via LLM Encoding and Pseudo Event Regulation for Video Moment Retrieval"
_acmmm.org/ACMMM/2024/Conference — MM2024 Oral_

### Official Review · Reviewer_6HgZ · 2024-05-08

**Rating:** 4
**Confidence:** 2

**Summary:**

This article proposes a new method to enhance Video Moment Retrieval (VMR) by utilizing a Large Language Model (LLM) encoder and pseudo event tuning. The research motivation stems from the limitations of LLM as a decoder for generating discrete text descriptions, which limits their direct application in generating continuous outputs. To overcome these limitations, the article suggests using LLM encoders instead of decoders. Through feasibility studies, the author has demonstrated that the LLM encoder can effectively refine the inter conceptual relationships in multimodal embeddings even without training on text embedding. In addition, the article also demonstrates that the refinement ability of LLM encoders can be transferred to other embeddings (such as BLIP and T5), as long as these embeddings exhibit similar cross concept similarity patterns to CLIP embeddings. The article proposes a universal framework for integrating LLM encoders into existing VMR architectures, particularly within fusion modules. The ability of LLM encoders to refine conceptual relationships helps the model balance its understanding of foreground and background concepts, rather than solely focusing on visually dominant foreground concepts. In addition, the article introduces the concept of pseudo events obtained through event detection techniques to guide the alignment of prediction times within the event boundary, effectively avoiding interference from adjacent times. Through experimental verification, the author has demonstrated the effectiveness of the proposed method and achieved state-of-the-art performance in VMR tasks.

**Strengths:**

Framework: The article proposes an innovative approach to using LLM encoders for video clip retrieval tasks, rather than traditional decoder applications.
Multimodal fusion: Through the LLM encoder, the article successfully integrates text and visual information to enhance semantic understanding and representation learning of video content.
Pseudo event regulation: The concept of pseudo events is introduced as a prior to the distribution of video content, which helps to improve the accuracy of time prediction.
Universal Framework: The proposed framework is designed as a pluggable component that is compatible with various existing VMR methods, demonstrating good adaptability and versatility.

**Limitations:**

Computational cost: Although the article proposes using some layers of LLM encoders to reduce computational requirements, the computational cost of LLM is still relatively high, which may limit its application in resource constrained environments.
Dependent on specific models: The performance of a method may depend on a specific LLM model, which may require adjustment or retraining for different LLM models.
Selection of negative samples: When adjusting for pseudo events, the selection of negative samples (i.e. non event boundary moments) may affect model performance.

**Suitability:**

3

---

### Official Review · Reviewer_H7pY · 2024-05-08

**Rating:** 4
**Confidence:** 2

**Summary:**

## Paper Topic And Main Contributions:

The paper proposes a general framework for integrating LLM encoders into existing Video Moment Retrieval architectures, and integration of prior knowledge via pseudo-event detector module. Its aim for LLM encoder idea is to investigate whether the LLM encoder approach can be applied to multimodal embeddings.

"Integration of LLM encoders into the fusion module" and "using the pseudo-events, obtained through event detection techniques, as priors for the content distribution of videos" are the main contributions.

**Strengths:**

## Strengths

1. The explored Idea is highly new.

2. LLM Encoding and Pseudo Event Regulation Ideas are well-explored with different models.

3. LLMs are used as encoders to overcome limitations such as “direct application to continuous outputs like salience scores and inter-frame embeddings” while LLMs are used as decoders for generating discrete textual descriptions.

**Limitations:**

## Weakness:

1. Detailed proposed explorations have been shown in detail. However, the Idea is not convincing enough to accept it as novel but highly new. Therefore, the improvement and impact do not show a high increase for the community.

2. Different LLM models could also be used for implementation.

## Questions:
1. Which Llama versions were used for implementation?
2. If the Llama1 was used, is it possible to reimplement via Llama2-7B or Llama3-8B?
3. Have you ever tried any other different models to implement?

Typos Grammar Style And Presentation Improvements:

1. This paper is well-organized and written.

2. From my perspective, that would be better to mention the contribution part specifically as separate paragraphs or bullet points. That way helps the reader to follow the flow.

3. It is not that important but, there is one typo (line 286) that would be better if fixed.

**Suitability:**

3

---

### Official Review · Reviewer_fceY · 2024-05-15

**Rating:** 4
**Confidence:** 3

**Summary:**

The paper is about enhancing video moment retrieval (VMR) by integrating large language model (LLM) encoders and pseudo-event regulation. The authors propose a general framework for VMR and introduce two plug-in elements, LLM encoders and pseudo-event regulation, to enhance the retrieval of specific moments in videos. The LLM encoders refine multimodal embeddings and their inter-concept relations, while pseudo-events serve as temporal content distribution priors to align moment predictions with actual event boundaries. The proposed methods are designed to be compatible with existing VMR frameworks and have been shown to achieve state-of-the-art performance.

**Strengths:**

Novelty: The paper proposes a novel approach by utilizing large language models (LLMs) as encoders for refining inter-concept relations in multimodal embeddings. This approach is different from traditional methods that use LLMs as decoders for discrete outputs.

Technical correctness: The paper demonstrates the feasibility of using LLM encoders to refine inter-concept relations in multimodal embeddings. It explains how LLM encoders operate as transformers, taking concepts as input and generating refined embeddings as output. The paper also discusses the compatibility of LLM encoders with different embeddings and the effectiveness of using a subset of LLM encoder layers for relation refinement.

Adequate evaluation: The paper conducts experiments to validate the proposed approach. It evaluates the ability of LLM encoders to refine relations in different scenarios, including when textual embeddings are predominant and when non-textual embeddings are distorted. The results demonstrate the effectiveness of LLM encoders as reliable relation refiners.

Clarity: The paper provides clear explanations of the approach, the rationale behind it, and the experimental setup. It describes the integration of semantic refinement using LLM encoders and pseudo-event regulation as plug-in components in existing video moment retrieval methods. The paper also discusses the benefits of these components in enhancing the performance of five different video moment retrieval frameworks.

**Limitations:**

1. The authors need to explain more about "inter-concept similarity patterns".
2. If you refine relations, wil it lead to the deterioration of abilities whe discriminating similar concepts?
3. Are the conclusion and experiments in section 3 all based on the sampled 1,000 triplets?
4. The authors need to explain "In other words, as long as the input concepts maintain inter-concept similarities similar to those of the CLIP embeddings"

**Suitability:**

3

---

### Official Review · Reviewer_K9Su · 2024-05-24

**Rating:** 5
**Confidence:** 1

**Summary:**

This paper explores how to leverage LLMs to integrate prior knowledge and enhance the performance of video moment retrieval (VMR) models through pseudo-event regulation. Due to the limitations of existing LLMs in generating discrete textual descriptions, which makes them challenging to apply directly for generating continuous outputs, this paper proposes using LLM encoders instead of decoders to refine multimodal embeddings. The feasibility study demonstrates that LLM encoders can effectively refine concept relationships in multimodal embeddings even without being trained on textual embeddings. Additionally, the concept of pseudo-events is introduced to guide segment predictions within event boundaries, avoiding interference from adjacent segments. Experimental validation shows that the proposed method achieves state-of-the-art performance in VMR tasks.

**Strengths:**

The paper proposes a method that combines LLM encoders and pseudo-event regulation to refine concept relationships in multimodal embeddings. This method offers a new perspective for video moment retrieval (VMR) by leveraging the powerful capabilities of LLM encoders to improve video understanding.

**Limitations:**

1.	Efficiency Analysis: Due to the high computational complexity of LLM models, it is recommended to include discussions and analyses on the operational efficiency of the method, especially regarding computational overhead and processing time on large-scale datasets. This is crucial for evaluating the feasibility of practical applications.
2.	Illustration Issues: The explanation of Figure 5 is not clear enough, and the start time of the second picture in Figure 6 should be 00:56 instead of 56:00.
3. The motivation with LLMs is not clear. It seems that the LLM can be instead

**Suitability:**

3

---

### Meta-Review · Area_Chair_CWBR · 2024-07-03

**Recommendation:** Accept (Oral)
**Confidence:** 5

**Metareview:**

The final ratings of this paper are three weak accept and one accept. After rebuttal,  all reviewers are in agreement that the paper's strengths warrant acceptance, and the AC agrees.

---

### Meta-Review · Senior_Area_Chairs · 2024-07-10

**Recommendation:** Accept (Oral)
**Confidence:** 5

**Metareview:**

All the reviewers gave positive ratings and tend to accept the paper. SAC and AC agree with reviewers and recommend acceptance of the paper.